

# Prevalence and risk factors of computer vision syndrome—assessed in office workers by a validated questionnaire

Natalia Cantó-Sancho[1], Stefano Porru[2,3], Stefano Casati[4], Elena Ronda[5,6], Mar Seguí-Crespo[1,5] and Angela Carta[2,3]

[1] Department of Optics, Pharmacology and Anatomy, University of Alicante, San Vicente del Raspeig, Alicante, Spain
[2] Department of Diagnostics and Public Health, University of Verona, Verona, Véneto, Italy
[3] Mistral-Interuniversity Research Centre 'Integrated Models of Study for Health Protection and Prevention in Living and Working Environments', University of Brescia, Milano Bicocca and Verona, University of Verona, Verona, Véneto, Italy
[4] Eye Clinic, Department of Neurosciences, Biomedicine and Movement Sciences, University of Verona, Verona, Véneto, Italy
[5] Public Health Research Group, University of Alicante, San Vicente del Raspeig, Alicante, Spain
[6] Biomedical Research Networking Center for Epidemiology and Public Health (CIBERESP), Madrid, Madrid, Spain

Corresponding author
Natalia Cantó-Sancho,
natalia.canto@ua.es

## ABSTRACT

**Background**. Computer vision syndrome (CVS) is a common occupational health problem, but its clinical definition, prevalence and risk factors are not well defined. In general, non-validated diagnostic instruments have been used to assess its prevalence. For this reason, the aim of this study is to estimate the prevalence and potential risk factors for CVS using a validated questionnaire.

**Methods**. A cross-sectional study ($n = 238$) was carried out in Italian office workers using digital devices. All participants responded to an anamnesis, a digital exposure questionnaire, and the validated Italian version of the Computer Vision Syndrome Questionnaire. A battery of 3 ocular surface and tear ophthalmic tests (break-up time, BUT), Schirmer II and corneal staining) was performed.

**Results**. The mean age ($\pm$SD) was 45.55 (11.02) years, 64.3% were female. 71.4% wore glasses to work, whose design was monofocal (for distance) in 47.6%, monofocal (for near) in 26.5%, general progressive in 16.5% and occupational progressive in 8.8% of cases. 35.7% used digital devices >6 hours/day in the workplace. The prevalence of CVS was 67.2%. In the multivariate model, female sex (aOR: 3.17; 95% CI [1.75–5.73]), the use of digital devices >6 hours/day at workplace (aOR: 2.07; 95% CI [1.09–3.95]) and the use of optical correction at work (aOR: 2.69; 95% CI [1.43–5.08]) significantly increased the odds of CVS. Association was observed between presenting CVS and having abnormal BUT ($\chi 2 = 0.017$).

**Conclusions**. The prevalence of CVS in Italian office workers, especially among females, was high. Intensive use of digital devices at work (>6 hours/day) and the use of optical correction at work significantly increased the odds of CVS. There is an association between poor tear stability and CVS. Further research is needed on the influence of wearing optical correction on CVS. The use of a validated questionnaire in health surveillance of digital workers is strongly recommended.

# INTRODUCTION

The use of digital devices both in the workplace and during leisure time activities is almost universal in our society. Although the rate of digitization of Italian companies is still limited compared to other European countries, in January 2022 there were 50.85 million internet users in Italy and internet penetration rate stood at about 84.3% of the total population. In the coming years, on the push towards digitization imposed by the pandemic emergency, a further significant increase in the number of workers and the time of use of video display units (VDUs) can be expected (*Nurra & Tomeo, 2020*; *Kemp, 2022*). In parallel with the spread and intensification of the use of digital technologies, an increase in related health issues and in particular an increase in computer vision syndrome (CVS) is likely to be expected (*Salinas-Toro et al., 2022*). CVS, also referred to as digital eye strain includes a set of ocular symptoms (such as dryness, itching and burning), visual symptoms (such as blurred or double vision and eye strain), and even headache, which are produced by prolonged use of computers, tablets, e-readers, and cell phones, among others (*Coles-Brennan, Sulley & Young, 2019*). The most recent reviews published to date indicate that risk factors related to CVS are: prolonged exposure time to VDUs (although there are discrepancies as to the cut-off point at which to consider exposure as a possible risk factor for CVS), female sex, presence of refractive, accommodative or vergence anomalies, altered blinking patterns, work environment (excessive exposure to intense light, low humidity, use of air conditioning, smaller font size), incorrect working posture, closer working distance, or use of contact lenses (*Coles-Brennan, Sulley & Young, 2019*; *Chawla et al., 2019*; *Auffret et al., 2021*; *Talens-Estarelles et al., 2021*; *Kaur et al., 2022*; *Adane, Alamneh & Desta, 2022*; *Singh et al., 2022*). In addition, in another review from 2022, a previous history of eye disease and the use of spectacles are also indicated as risk factors (*Adane, Alamneh & Desta, 2022*). However, not all reviews report the same risk factors for CVS. For example, there is much controversy as to whether age is a risk factor for CVS (*Auffret et al., 2021*; *Talens-Estarelles et al., 2021*; *Rosenfield, 2011*). It should be noted that most of the studies referenced in the reviews use instruments that are not validated for the diagnosis of CVS.

Some studies that use validated diagnostic methods and have studied CVS risk factors have found that female sex is a risk factor, but for example, opposite results were obtained for exposure to VDUs (*Sánchez-Brau et al., 2020*; *Artime-Ríos et al., 2022*). In addition, the study by *Sánchez-Brau et al. (2020)*, found that non-neutral neck position and altered workplace lighting were risk factors, while no statistical significance was found for age, altered humidity or prolonged exposure to VDUs. Because of this disparity between research, and as also indicated by *Kaur et al. (2022)*, "future studies should focus on understanding the risk factors among different groups", and to this end, use validated instruments for the diagnosis of CVS, especially because its pathophysiology is still poorly understood. This seems to be multi-factorial and includes changes in the balance of

accommodation-convergence and ocular surface changes. As far as accommodation and convergence is concerned, these studies are mainly old and the results are heterogeneous (*Collier & Rosenfield, 2011*; *Qu et al., 2005*; *Sheppard & Wolffsohn, 2018*), and the most recent ones focus on the children/teenager student population (*De Hita Cantalejo et al., 2021*; *Mohan et al., 2021*; *De Hita Cantalejo et al., 2022*), or seek to evaluate treatments but not causes of CVS (*Seguí-Crespo et al., 2022*). To the contrary, in recent years, many studies have shown an increase in dry eye signs and symptoms of dry eye disease in VDU workers (*Talens-Estarelles et al., 2021*; *Mehra & Galor, 2020*; *Sánchez-Valerio et al., 2020*), probably due to reduced blink rate or incomplete blinks (*Portello, Rosenfield & Chu, 2013*). Although there is a large body of research on CVS, the extreme heterogeneity of the methods of measuring outcomes, in particular the definition of CVS, limits the interpretation of the results and the possibility of evaluating the effectiveness of any preventive measures. This is highlighted in a recent systematic review carried out by the Cochrane Collaboration about efficacy of optical correction of refractive error for preventing and treating eye symptoms in computer users (*Heus, Verbeek & Tikka, 2018*). The diagnosis of the CVS is often based on non-validated questionnaires, which include a different set of symptoms depending on the author's criteria, and with an imprecise definition of when a worker should be considered symptomatic (*Coles-Brennan, Sulley & Young, 2019*). Therefore, critical issues are inaccurate definitions of when a person should be considered symptomatic and the focus on the syndrome as a global issue rather than on different individual symptoms.

One of the most recently published narrative reviews found that the prevalence of CVS is estimated to range from 25 to 93% in general population (*Coles-Brennan, Sulley & Young, 2019*). This range is so wide as it depends on the cohort of the population studied, the definition of CVS and the methodology employed to measure it (*Coles-Brennan, Sulley & Young, 2019*). Specifically, a prevalence of CVS has been reported to be around 70% among VDU workers (*Sánchez-Brau et al., 2020*; *Sánchez-Valerio et al., 2020*). Particularly in Italy, studies report a prevalence of CVS in workers ranging from 13.3% to 88.6% (*Carta et al., 2003*; *Fenga et al., 2005*; *Taino et al., 2006*; *Fenga et al., 2007*; *Fenga et al., 2008*; *Carta et al., 2010*; *Larese Filon et al., 2019*). In all of these studies symptoms were collected using ad hoc questionnaires or an unvalidated standardized questionnaire. All questionnaires must demonstrate their validity, reliability, and responsiveness, and all these properties are determined in the instrument validation process. Thus, the use of non-validated questionnaires does not guarantee the quality of the results obtained, and the conclusions drawn may be meaningless, inappropriate and not corroborated by the results (*Ramada-Rodilla, Serra-Pujadas & Delclós-Clanchet, 2013*; *Carvajal et al., 2011*; *Leong et al., 2016*).

In 2015, a validated patient reported outcome (PRO) questionnaire was developed in Spain with good psychometric properties to diagnose CVS, the Computer Vision Syndrome Questionnaire (CVS-Q©) (*Seguí-Crespo et al., 2015*). A linguistic version of the CVS-Q© was translated, cross-culturally adapted, and validated into Italian, the CVS-Q IT© (*Seguí-Crespo et al., 2019*; *Cantó-Sancho et al., 2022*). The aim of this study is to estimate the prevalence of CVS and its relationship with potential risk factors in a sample of Italian VDU users using this validated questionnaire.

## MATERIALS & METHODS

A cross-sectional epidemiological study was carried out in Italian office workers using VDUs from May to July 2019. Workers were included if they used the following VDUs at work: mobile phones, laptops, computers, tablets, or e-readers. Workers aged 18–67 years, working full time during day, 5 days a week, and who regularly used more than 2 h digital devices during their working day were recruited by the Occupational Medicine Unit at the University Hospital of Verona (Italy) during the regular and mandatory health surveillance activities. All participants included were native Italian and they signed the informed consent form. Workers who wore contact lenses daily (contact lenses users who will use them on an occasional basis for one-off activities were included), who were undergoing refractive or cataract surgery, suffering from any ocular pathology and/or undergoing ocular treatment (including regular use of artificial tears) in the 3 months prior to the study, which could affect CVS symptomatology, were excluded. To calculate the sample size, the calculator GRANMO version 7.12 was used. Considering that the *Azienda Ospedaliera Universitaria Integrata Verona* (based in Borgo Roma) is composed of approximately 2,500 workers in total who use VDUs. It was calculated that a sample of $n = 209$ workers was enough to estimate with a confidence level of 90% and an accuracy of $\pm 5$ percentage units and a predictable population percentage of 70% (*Sánchez-Brau et al., 2020*).

For each participant, an anamnesis was collected, taking into account (1) sociodemographic information, (2) general health, (3) ocular health, (4) optical correction information, (5) variables of exposure to digital devices, and (6) use of air conditioning at work. All these variables were self-reported by the worker. Furthermore, CVS symptoms using CVS-Q IT© were collected and a battery of three ocular surface and tear ophthalmic tests were performed in both eyes (see Appendix 1 for more information on the variables studied).

The CVS-Q IT© is a scale that evaluates the frequency (never: the symptom does not occur at all, occasionally: sporadic episodes or once a week, and often or always: 2 or 3 times a week to almost every day) and intensity (moderate or intense) of 16 ocular and visual symptoms related to the digital devices use. The questionnaire instructions ask the worker to indicate whether he/she experiences any of the following symptoms during the time he/she uses the computer at work. Subsequently the frequency and intensity data are recoded to calculate the severity of each symptom, resulting in a total score. Total scores $\geq 7$ indicate that the subject suffers CVS (*Cantó-Sancho et al., 2022*).

Tear stability (break-up time, BUT), presence of corneal staining, and tear quantity (Schirmer II) were evaluated in the following order. The clinical tests were carried out using the slit lamp, fluorescein strips, blue filter, ocular anaesthetic and Schirmer's absorbent paper strips. For the three clinical tests, the normality criteria established by the TFOS DEWS II Report were followed (*Wolffsohn et al., 2017*), which considers BUT to be abnormal when it is $\leq 10$ s, the existence of >5 staining points to be considered abnormal evidence, and the tear quantity to be inadequate when the wet part of the absorbent strip is $\leq 10$ mm, after 5 min. As the clinical tests were performed on both eyes, to classify the test as abnormal/normal, data from a single randomly selected eye were considered.

Clinical data, CVS-Q IT© and clinical tests were carried out by experienced staff who agreed on the clinical criteria to adopt. An optician-optometrist (NCS) explained the study, gave informed consent and collected the anamnesis and CVS-Q IT© data. Ophthalmologists (SC) performed the battery of clinical tests.

All the study was conducted following the standards of Good Clinical Practice and international ethical principles applicable to human research, according to the latest revision of the Declaration of Helsinki. The study was approved by the Ethics Committee of the University of Alicante (UA-2018-02-22) and by the Comitato Etico per la Sperimentazione Clinica delle Province di Verona e Rovigo (41605).

### Statistical analysis

A descriptive analysis of all study variables was performed. Absolute frequency and percentage were calculated for categorical variables. For continuous variables, the mean and standard deviation (SD) and the minimum and maximum were obtained. The frequency of the 16 symptoms included in the questionnaire was calculated, as well as the total prevalence of CVS and for each variable and category; differences between groups were assessed using $\chi 2$ test. Additionally, logistic regression models were calculated to measure the association between CVS and the remaining variables. The crude odds ratios (cOR) and adjusted odds ratios (aOR) were calculated plus 95% confidence interval (95% CI). A $p$-value of less than 0.05 was considered statistically significant. The statistical software SPSS version 28 was used for the analysis.

## RESULTS

A total of 296 workers participated in the study, but a total of 19.6% were excluded for different reasons: the presence of an ocular pathology or a pharmacological treatment at the time of the study being the most frequent reason (53.4%). The final sample amounted to a total of $n = 238$ participants. The mean age ($\pm$SD) was 45.55 (11.02) years with a range between 23 and 67 years (the median age was 48 years); 64.3% were female. 9.7% presented with a past history of ocular disorders (mainly conjunctivitis and strabismus during childhood). 4.2% had prior ocular surgery (mainly chalazion) and 2.5% rarely used artificial tears as ocular pharmacological treatment. 79.0% used optical correction regularly, 16.8% of them also used contact lenses sporadically (sports or weekend), never daily or for work. Specifically, 71.4% wore glasses to work, whose design was monofocal (for distance) in 34.0% (47.6% if only glasses wearers are considered), monofocal (for near) in 18.9% (26.5% if only glasses wearers are considered), general progressive in 11.8% (16.5% if only glasses wearers are considered) and occupational progressive in 6.3% (8.8% if only glasses wearers are considered) of cases. Only 1 person used glasses with a bifocal design and was therefore excluded from the prevalence and association analysis. 56.7% of the workers indicated that they were presbyopic, although of these, 46 did not use correction for work, perhaps because they were nearsighted. 35.7% of the sample used digital devices >6 hours/day in the workplace; during the day, the average time spent using digital devices at work was 5.85 (1.54) hours/day, with a range between 2 and 10 hours/day. Most of the participants (79.4%) could be considered VDU workers according to Italian regulations as

they worked more than 20 h a week with digital devices. The remaining (20.6%) were also VDU users but did not use them for more than 20 h a week. A total of 30.3% used digital devices >8 hours/day including work and leisure, with a range between 3 and 17 hours/day (Table 1).

The total prevalence of CVS was 67.2% and the mean CVS-Q IT$^{©}$ score was 7.08 (3.91) points. The mean CVS-Q IT$^{©}$ score for those without CVS was 2.83 (1.65) points and the mean CVS score for those with CVS was 9.16 (2.87) points. Statistically significant differences were observed between the mean CVS-Q IT$^{©}$ score in people with and without CVS ($p < 0.001$). In addition, statistically significant differences in prevalence were observed by sex ($p < 0.001$), regular use of optical correction ($p < 0.001$) and for work ($p < 0.001$), and to the time of use of digital devices to work ($p = 0.010$) (Table 1).

The most frequent symptoms were blurred vision (63.5%), feeling that sight is worsening (61.8%), headache (56.3%), and burning (54.2%) and the least were eye pain (11.3%), coloured halos around objects (16.4%), and double vision (17.6%); almost all symptoms occurred occasionally in most cases. However, blurred vision and difficulty focusing for near vision were often or always present in 16% of the sample analyzed. All participants felt symptoms with moderate intensity (Fig. 1).

Significant associations were found with female sex (cOR: 3.42; IC 95% [1.94–6.04]), use of optical correction on a regular basis (cOR: 3.13; CI 95% [1.65–5.95]) and for work (cOR: 3.11; CI 95% [1.73–5.60]). An association between lens design and CVS was apparent for monofocal (distance) (cOR: 3.71; CI 95% [1.83–7.53]), general progressive lenses (cOR: 3.18; CI 95% [1.20–8.47]) and occupational progressive lenses (cOR: 4.24; CI 95% [1.10–10.07]). Regarding digital devices use, the association with CVS was higher among workers who used digital devices >6 hours/day to work (cOR: 2.20; CI 95%: [1.20–4.04]) and among those who used them >8 hours/day in total (for work and leisure purposes) (cOR: 2.19; CI 95% [1.06–4.54]) compared to those who used them ≤6 hours/day in total (Fig. 2). After adjusting for sex, age, optical correction for work and hours of use of digital devices at work, the results of the multivariate analysis indicated three factors that were associated with CVS in the proposed model. The use of digital devices >6 hours/day at work (aOR: 2.07; 95% CI [1.09–3.95]) doubles the odds of suffering from CVS, while being female (aOR: 3.17; 95% CI [1.75–5.73]) and the use of optical correction at work (aOR: 2.69; 95% CI [1.43–5.08]) triples the odds to suffer from it (Fig. 2). Note that Fig. 2 shows the results for those variables with statistically significant results of the simple logistic regression (See Appendix 2 for comprehensive analysis).

Regarding the battery of ocular surface and tear tests, Table 2 shows the values of the clinical tests for the right and left eye, as well as for the random choice, which will allow us to classify the tests as normal or abnormal. Following the criteria recommended by the TFOS DEWS II (Wolffsohn et al., 2017), we observed that 19.3% of the workers had corneal staining (>5 staining points), 57.1% had poor tear stability (abnormal BUT ≤10 s) and 45.8% of the sample did not have good tear quantity (abnormal Schirmer II ≤10 mm). Association was observed between presenting CVS and having abnormal BUT ($\chi 2 = 0.017$), and a small but significant Pearson correlation is obtained (r = −0.143,
**Table 1  Distribution of the studied sample (*n* = 238), prevalence of computer vision syndrome (CVS) and differences according to sociodemographic characteristics, eye health, optical correction, and exposure to digital devices variables.**

| | Full population | | CVS population | | |
|---|---|---|---|---|---|
| | N | % | N | prevalence (%)[§] | *p*-value |
| **Total** | **238** | **100** | **160** | **67.2** | |
| **Sex** | | | | | |
| Male | 85 | 35.7 | 42 | 49.4 | < 0.001[***] |
| Female | 153 | 64.3 | 117 | 77.0 | |
| **Age (years)** | | | | | |
| ≤ 40 | 76 | 31.9 | 48 | 63.2 | 0.376 |
| > 40 | 162 | 68.1 | 111 | 68.9 | |
| **Workplace** | | | | | |
| University of Verona | 195 | 81.9 | 129 | 66.5 | 0.679 |
| Hospital Borgo Roma | 43 | 18.1 | 30 | 69.8 | |
| **General pharmacological treatment** | | | | | |
| No | 149 | 62.6 | 98 | 65.8 | 0.575 |
| Yes | 89 | 37.4 | 61 | 69.3 | |
| **Past ocular disorders** | | | | | |
| No | 215 | 90.3 | 145 | 67.8 | 0.504 |
| Yes | 23 | 9.7 | 14 | 60.9 | |
| **Ocular surgery[†]** | | | | | |
| No | 228 | 95.8 | 153 | 67.4 | 0.733 |
| Yes | 10 | 4.2 | 6 | 60.0 | |
| **Ocular pharmacological treatment[†]** | | | | | |
| No | 232 | 97.5 | 153 | 66.2 | 0.182 |
| Yes | 6 | 2.5 | 6 | 100.0 | |
| **Regular optical correction** | | | | | |
| No | 50 | 21.0 | 23 | 46.0 | < 0.001[***] |
| Yes | 188 | 79.0 | 136 | 72.7 | |
| **Use of glasses to work** | | | | | |
| No | 68 | 28.6 | 33 | 48.5 | < 0.001[***] |
| Yes | 170 | 71.4 | 126 | 74.6 | |
| **Lens design at work[‡]** | | | | | |
| Nothing | 68 | 28.6 | 33 | 48.5 | |
| Bifocal | 1 | 0.4 | – | – | |
| Monofocal distance | 81 | 34.0 | 63 | 77.8 | 0.002[**#] |
| Monofocal near | 45 | 18.9 | 30 | 66.7 | |
| General progressive | 28 | 11.8 | 21 | 75.0 | |
| Occupational progressive | 15 | 6.3 | 12 | 80.0 | |
| **Presbyopia** | | | | | |
| No | 103 | 43.3 | 67 | 65.0 | 0.558 |
| Yes | 135 | 56.7 | 92 | 68.7 | |

*(continued on next page)*

| | Full population | | CVS population | | |
|---|---|---|---|---|---|
| | N | % | N | prevalence (%)$^{\S}$ | p-value |
| **Occupational use of digital devices (hours/day)** | | | | | |
| ≤ 6 | 153 | 64.3 | 93 | 61.2 | 0.010* |
| > 6 | 85 | 35.7 | 66 | 77.6 | |
| **Years working with digital devices** | | | | | |
| ≤ 10 | 72 | 30.3 | 45 | 62.5 | |
| 11–20 | 93 | 39.1 | 64 | 68.8 | 0.609 |
| > 20 | 73 | 30.6 | 50 | 69.4 | |
| **Scheduled breaks during work with digital devices** | | | | | |
| No | 34 | 14.3 | 24 | 70.6 | 0.639 |
| Yes | 204 | 85.7 | 135 | 66.5 | |
| **Duration of breaks (minutes)** | | | | | |
| ≤ 5 | 87 | 36.5 | 58 | 66.7 | |
| 6–10 | 98 | 41.2 | 67 | 69.1 | 0.824 |
| > 10 | 53 | 22.3 | 34 | 64.2 | |
| **Use of air conditioning at work** | | | | | |
| Never or rarely | 36 | 15.1 | 20 | 55.6 | 0.110 |
| Often or always | 202 | 84.9 | 139 | 69.2 | |
| **Use of digital devices for leisure (hours/day)** | | | | | |
| ≤ 2 | 191 | 80.3 | 127 | 66.8 | 0.871 |
| > 2 | 47 | 19.7 | 32 | 68.1 | |
| **Total use of digital devices (hours/day)** | | | | | |
| ≤ 6 | 65 | 27.3 | 37 | 57.8 | |
| 6–8 | 101 | 42.4 | 68 | 67.3 | 0.103 |
| > 8 | 72 | 30.3 | 54 | 75.0 | |

**Notes.**

[†] The $\chi^2$ test has been used for all the variables, except for these two variables that Fisher's exact test has been used.

[‡] The category "bifocal" of these variable has been excluded from the analysis of the prevalence since there was only one person.

[*] p-value < 0.05.

[**] p-value < 0.01.

[***] p-value < 0.001.

[§] Formula for prevalence: persons presenting the event/total number of persons in that category.

[#] Since the prevalences between the different lens' designs are very similar, this statistical significance is not due to the lens' design itself, but to the fact of wearing or not glasses to work.

$p = 0.027$). No significant association was observed between presenting CVS and having abnormal Schirmer II or corneal staining (Table 2).

# DISCUSSION

The results of this study detected a prevalence of CVS of around 70% in Italian office workers. Blurred vision, feeling that sight is worsening, headache and burning as the most frequent symptoms. After adjusting for sex, age, optical correction for work and hours of use of digital devices at work, the increase in CVS is explained by three factors; female sex, use of optical correction to work and use of digital devices >6 hours/day at work. Furthermore, a negative and significant association between CVS and tear stability was also observed.
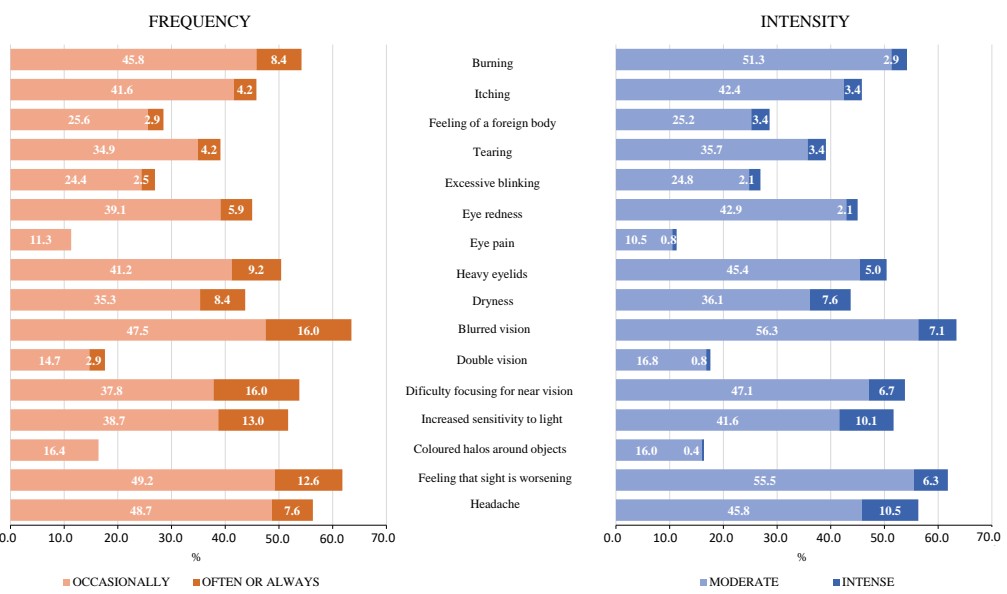

**Figure 1** Frequency and intensity with which workers perceive each of the 16 symptoms that make up the CVS-Q IT©.

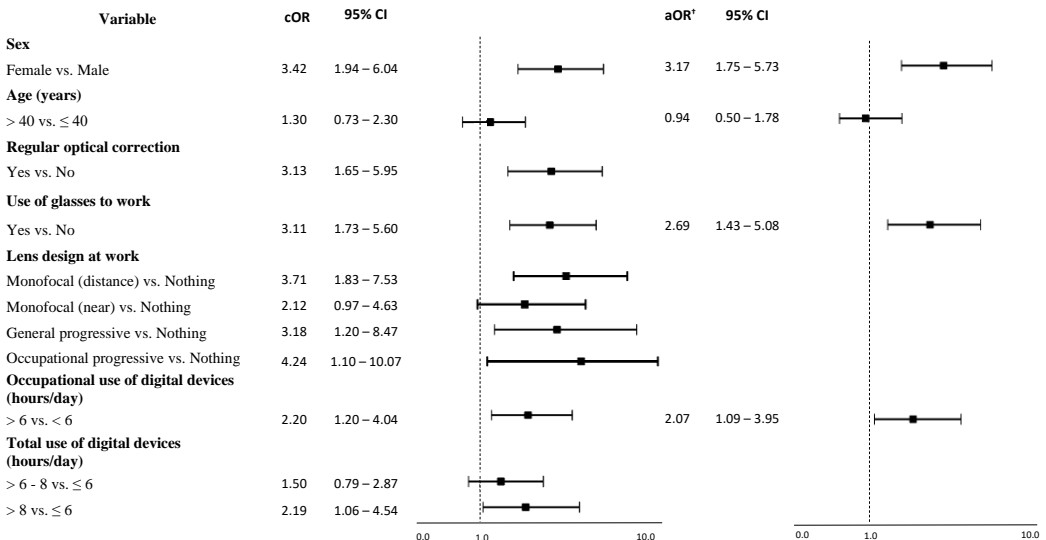

† The model has been adjusted for the variables: sex, age, optical correction for work and digital devices use for work.

**Figure 2** Association between computer vision syndrome and variables studied: crude odds ratios (cOR) and adjusted odds ratios (aOR) and 95% confidence intervals (95% CI).

Our results cannot be directly compared with previous research on Italian workers, because non-validated questionnaires were used in other studies for the diagnosis of CVS in the Italian population and, except the one by *Larese Filon et al. (2019)*, these were published more than 10 years ago when the equipment were likely different (*Taino et al.,*

**Table 2  Results of ocular surface and tear ophthalmic tests of the sample analysed (BUT, Schirmer II and corneal staining) and their association with computer vision syndrome (CVS).**

|  | BUT (s) | Schirmer II (mm) | Corneal staining (points) |
|---|---|---|---|
| RE (mean ± SD) | 9.91 ± 4.44 | 13.02 ± 7.86 | - |
| LE (mean ± SD) | 9.88 ± 4.47 | 12.97 ± 8.13 | - |
| Random selected eye (mean ± SD)[†] | 10.17 ± 4.64 | 12.99 ± 8.10 | - |
| Abnormal test, n (%) | 136 (57.1) | 109 (45.8) | 46 (19.3) |
| No CVS, n (%) | 36 (46.2) | 31 (39.7) | 13 (16.7) |
| CVS, n (%) | 100 (62.5) | 78 (48.8) | 33 (20.6) |
| $p$-value[‡] | 0.017[*] | 0.191 | 0.468 |

**Notes.**

RE, right eye; LE, left eye.

[†] These are the results of the random selection of the eye for each participant (which will allow us to classify the test as normal or abnormal).

[-] is not a continuous variable.

[‡] $\chi^2$ test.

[*] $p$-value $< 0.05$.

2006; *Carta et al., 2010*). The prevalence observed in this study (67.2%) is higher than that reported in Italian VDU workers, except when compared to the studies by *Fenga et al. (2005)* (prevalence = 79.0%) and *Fenga et al. (2008)* (prevalence = 88.6%), though in these two researches, samples of respectively 54 and 70 VDU workers were studied. In general, the target population studied when conducting research on the prevalence of this syndrome in the Italian population, tends to be administrative workers belonging to public offices, service agencies or to hospital/university (just as in our case) (*Carta et al., 2003*; *Fenga et al., 2005*; *Fenga et al., 2008*; *Carta et al., 2010*; *Larese Filon et al., 2019*). However, given that the other studies have not used CVS diagnostic instruments that have been shown to be reliable and valid, the quality of their results cannot be guaranteed. Since this study has used an instrument that has previously been culturally adapted and validated for the Italian population (*Seguí-Crespo et al., 2019*; *Cantó-Sancho et al., 2022*), in future studies, it would be advisable to use the data obtained in this study for comparisons. The observed prevalence is high, similar to a pilot study in which CVS-Q IT© was also used (*Seguí-Crespo et al., 2019*) and within the range of rates observed in other studies that have also used the CVS-Q© to study CVS prevalence in Spanish workers, that is 53.0–74.3% (*Sánchez-Brau et al., 2020*; *Artime-Ríos et al., 2022*; *Sánchez-Brau et al., 2021*; *Tauste et al., 2016*).

The high prevalence of symptoms observed in females (female: 77.0% *vs* male: 49.4%), is in line with the literature (*Carta et al., 2010*; *Portello et al., 2012*; *Toomingas et al., 2014*; *Ranasinghe et al., 2016*). However, in the study by *Larese Filon et al. (2019)*, which also analyzed an Italian sample, no association was found between being female and visual fatigue. This may be because the authors consider only visual fatigue (as a single symptom) and did not take CVS as a global construct, which considers other symptoms of the ocular surface such as dryness. In addition, they evaluated the frequency of eye symptoms related to digital device use, but they did not indicate which instrument they used, and whether it was validated or not. In our study we have found that being female triples the odds of suffering from CVS, in other studies who also use the CVS-Q© it has been observed

that it doubles (OR: 2.0; 95% CI [1.36–2.95]) (*Tauste et al., 2016*) or triples (OR: 3.40; 95% CI [1.12–10.33], OR: 2.85; 95% CI [1.03–7.83] and OR: 2.57; 95% CI [1.36–4.88]) (*Sánchez-Brau et al., 2020*; *Artime-Ríos et al., 2022*; *Zayed et al., 2021*). In any case, there seems to be a clear association between being female and a greater probability of suffering from visual and ocular symptoms or even CVS (*Sánchez-Brau et al., 2020*; *Courtin et al., 2016*; *Sullivan et al., 2017*). In addition, our study observed that females were more likely to fail two of the three ocular surface and tear tests than males (45.1% *vs.* 27.1%, $p = 0.006$), and that there were significant differences in the mean BUT according to sex (female = $9.69 \pm 4.44$s *vs.* male = $11.04 \pm 4.90$s, $p = 0.032$). Considering that there are no significant differences between males and females, neither in terms of age nor in terms of the average time of use of VDU for work or in total (work and leisure), these sex differences according to TFOS DEWS II could be due to the ''effects of sex steroids (*e.g.*, androgens, estrogens), hypothalamic-pituitary hormones, glucocorticoids, insulin, insulin-like growth factor 1 and thyroid hormones, as well as to the sex chromosome complement, sex-specific autosomal factors and epigenetics (*e.g.*, microRNAs)'' (*Sullivan et al., 2017*).

An association between intensive use of digital devices at work and a greater probability of suffering from ocular and visual symptoms or CVS ($\leq 6$ hours/day: 61.2% *vs* >6 hours/day: 77.6%) has also been reported in the scientific literature (*Talens-Estarelles et al., 2021*; *Larese Filon et al., 2019*; *Robertson, Huang & Larson, 2016*). However, some authors found statistical significance with a cut-off point of 2-4 or 4 hours/day (*Artime-Ríos et al., 2022*), others with 6 hours/day (like us) (*Tauste et al., 2016*; *Zayed et al., 2021*; *Raja et al., 2015*), 7 hours/day (*Rahman & Sanip, 2011*), 8 hours/day (*Uchino et al., 2013*), or even only analyze the correlation (as a continuous variable) (*Ranasinghe et al., 2016*); making comparison between studies difficult. It would therefore be useful to determine from how many hours of work with digital devices exposure could be considered as a risk factor for CVS. In our study, we have found that working >6 hours/day with digital devices duplicates the probability of suffering from CVS and other research also confirms that working >6 hours/day with digital devices is linked to an increased risk of CVS (*Tauste et al., 2016*; *Zayed et al., 2021*; *Raja et al., 2015*). It should be noted, however, that those who declare ''more than 6 h per day'' probably do not respect the legislative indication relating to the way in which breaks are used and the organization of work (in fact, the Italian law–Legislative Decree n.81/2008—prescribes 15 min breaks from VDU works every 120 min of VDU works). On the other hand, this result suggests that the indications of the legislation are widely protective for exposed workers with respect to the occurrence of CVS.

The higher probability of suffering from ocular and visual symptoms or CVS in workers using optical correction at work agreed with some literature but not with other studies. *Bhanderi, Choudhary & Doshi (2008)* found that subjects who have refractive errors (even when corrected) are more likely to develop CVS. Similarly, *Artime-Ríos et al. (2022)*, detected a significantly higher prevalence of CVS in ophthalmic lens wearers (OR: 1.88; 95% CI [1.22–2.89]), and *Zayed et al. (2021)* showed a significant relationship between CVS and eyeglass wear (OR: 5.01; 95% CI [1.09–23.06]), but neither of them specified whether it is on a regular basis or just for work. On the other hand, neither *Carta et al. (2003)* nor *Sánchez-Brau et al. (2020)* found a significant association between having a refractive

error (which can be corrected with ophthalmic lenses) and an increased probability of suffering from CVS. In a time series study with a quasi-experimental design (*Sánchez-Brau et al., 2021*), in which subjects were prescribed first a general progressive and then an occupational lenses (both with the optimal prescription for the worker), it was observed that workers had lower CVS (assessed with the CVS-Q$^{©}$), when using occupational lenses. Therefore, if the improvement in CVS is only due to proper worker correction, the same improvements should be observed with general and occupational progressive lenses. Furthermore, it was also observed that ametropic workers (even if well corrected) are less likely to improve CVS symptoms when switching from general progressive to occupational lenses. On the other hand, in our study, in which neither refraction nor visual acuity tests were performed, it could be thought that the higher frequency of visual symptoms, such as blurred vision or feeling that sight is worsening, could be because subjects did not have adequate vision. However, in other studies, which have included only "individuals with corrected binocular visual acuity, at far and near distance, to at least 0.0 logMAR", and who have also used the CVS-Q$^{©}$ as an instrument for CVS diagnosis, they have also found that the most frequent symptoms of CVS are: difficulty focusing for near vision, feeling that sight is worsening and blurred vision (*Sánchez-Brau et al., 2020*; *Sánchez-Brau et al., 2021*). Therefore, more research is necessary to explain the effects of wearing optical correction on CVS, especially if measures are incorporated to evaluate uncorrected refractive errors (*Sheppard & Wolffsohn, 2018*; *Rosenfield et al., 2012*). Some possible explanations for the increased probability of CVS among those workers using optical correction in our research are: (1) the group composed of optical correction users are not well corrected (*Sheppard & Wolffsohn, 2018*), (2) the presbyopic workers do not present adequate correction for near vision, and this causes their neck posture during VDUs work to be inadequate (*Sánchez-Brau et al., 2020*), (3) if the worker has subclinical accommodative/vergence difficulties, even if the worker is well corrected for the refractive problem (*i.e.,* myopia, hyperopia, astigmatism or presbyopia), the prolonged and demanding near work (involving higher visual demands) and constant refocusing at different working distances (looking at the screen, paper documents and the keyboard at the same time) (*Yan et al., 2008*), can cause these problems to surface during long hours of close-up work (*Sheppard & Wolffsohn, 2018*). All of these possibilities could cause an increased probability of CVS, but could be prevented with a complete refractive and binocular examination by the specialist. There is also a possibility that these differences are because people with CVS or ocular/visual symptomatology are more likely to have their eyes checked and to be prescribed optical correction by specialists. All this should be further investigated in future studies.

Finally, regarding the results of the objective tests carried out, a recent review on ocular surface alterations concluded that there is a reduction in tear volume, a noticeable decrease in tear stability and alterations in tear film composition among digital device users. Signs may appear due to incomplete blinking, resulting from increased cognitive and task demands while working with digital devices (*Talens-Estarelles et al., 2021*). In our results, around 50% of the workers had alterations in both tear volume and stability, which is consistent with studies in the literature. There is a discrepancy between the results of studies as to whether workers who use digital devices have more corneal staining than

those who do not work with digital devices (*Talens-Estarelles et al., 2021*). In our case the frequency of workers with corneal staining did not reach 20%. In another study, also among Italian office workers, no relationship has been observed between having an abnormal Schirmer test and BUT and having CVS (*Carta et al., 2003*).

Regarding the limitations of the present study, one is that the non-inclusion of workers who habitually used contact lenses to work may have skewed our sample to focus on an older population (*e.g.*, there are no subjects between 18 and 23 years). However, this was chosen because it has been widely seen in the literature that contact lens wear impacts normal ocular surface homeostasis, which may cause users to report more ocular discomfort (such as dryness, irritation, *etc.*) (*Nichols et al., 2013*; *Stapleton et al., 2017*) and in this study we did not want the prevalence of CVS to be affected by this condition. Another limitation is that objective tests could have been carried out to assess the refractive state of workers, since uncorrected refractive errors can increase symptoms related to digital devices (*Rosenfield, 2011*; *Sheppard & Wolffsohn, 2018*). However, this study was intended to present a more observational cross-sectional design, and therefore only changes in the ocular surface were evaluated by objective ocular surface and tear tests. A further limitation of this study, shared however with most studies on this aspect, is the absence of objective data related to the real use of the devices (time, type and mode of use) and data on the environmental conditions that could facilitate the appearance of disturbances (indoor air quality, lighting conditions). Regardless of all, this is the first study in the Italian working population in which a validated instrument was used to estimate the prevalence of CVS and some of its risk factors, which is its main strength. However, it should be noted that this is a cross-sectional study, so just correlation, but no causality, can be established. The frequency and intensity of ocular and visual symptoms associated with the intensive use of digital devices is expected to reach increasingly higher values, especially in the working population, partly due to the pandemic and telework (*Salinas-Toro et al., 2022*). Therefore, previous prevalence data with validated instruments are needed to assess the impact that teleworking can have on the visual and ocular health of the general population, and specifically in VDU workers. Comprehensive research including the collection of objective data on environmental factors that could influence the onset of CVS could make an important contribution to the planning of preventive interventions. In particular, the use of validated questionnaires, as suggested in a recent review of the Cochrane collaboration literature, could make an important contribution to the evaluation of the effectiveness of preventive interventions (*Heus, Verbeek & Tikka, 2018*). The use of a validated questionnaire in health surveillance of digital workers is strongly recommended.

## CONCLUSIONS

In conclusion, the prevalence of CVS in Italian office workers, especially among females, was high. Intensive use of digital devices at work and the use of optical correction to work significantly increased the probability of CVS. A significant association has been found between poor tear stability and CVS. It is essential to continue to investigate the influence of wearing optical correction on CVS in the context of multidisciplinary research using

validated instruments as well as rigorous data collection and analysis methodologies. A better understanding of the syndrome will allow us to determine the preventive measures needed to reduce it. The Italian version of the CVS-Q$^{©}$ (the CVS-Q IT$^{©}$) is a useful and valid tool for health surveillance of digital devices workers and efforts should be made to include validated questionnaires in daily preventive and clinical activities. Finally, a validate tool would potentially be available for multicenter, prospective cohort studies, which could clarify the still open research questions regarding CVS.

## ACKNOWLEDGEMENTS

All authors would like to thank the resident in ophthalmology Dra. Manuela Mambretti for her contribution in performing the clinical tests. This article will form part of the first author's doctoral thesis.

### Funding

The Vice-Rectorate of Research of the University of Alicante funded the pre-doctoral training contract for Natalia Cantó-Sancho (UAFPU2019-08). The funders had no role in study design, data collection and analysis, decision to publish, or preparation of the manuscript.

### Grant Disclosures

The following grant information was disclosed by the authors:
The Vice-Rectorate of Research of the University: UAFPU2019-08.

### Competing Interests

The authors declare there are no competing interests.

### Author Contributions

- Natalia Cantó-Sancho conceived and designed the experiments, performed the experiments, analyzed the data, prepared figures and/or tables, authored or reviewed drafts of the article, and approved the final draft.
- Stefano Porru conceived and designed the experiments, authored or reviewed drafts of the article, and approved the final draft.
- Stefano Casati performed the experiments, authored or reviewed drafts of the article, and approved the final draft.
- Elena Ronda conceived and designed the experiments, authored or reviewed drafts of the article, and approved the final draft.
- Mar Seguí-Crespo conceived and designed the experiments, analyzed the data, authored or reviewed drafts of the article, and approved the final draft.
- Angela Carta conceived and designed the experiments, authored or reviewed drafts of the article, and approved the final draft.

## Human Ethics

The following information was supplied relating to ethical approvals (i.e., approving body and any reference numbers):

All the study was conducted following the standards of Good Clinical Practice and international ethical principles applicable to human research, according to the latest revision of the Declaration of Helsinki. The study was approved by the Ethics Committee of the University of Alicante (UA-2018-02-22) and by the Comitato Etico per la Sperimentazione Clinica delle Province di Verona e Rovigo (41605).

## Data Availability

The data used in the analysis are available in the Supplemental File.

## Supplemental Information

Supplemental information for this article can be found online at http://dx.doi.org/10.7717/peerj.14937#supplemental-information.

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
