# Peer review of "Prevalence and risk factors of computer vision syndrome—assessed in office workers by a validated questionnaire"

_PeerJ, doi:10.7717/peerj.14937_

## Round 0.1 · original submission · Minor Revisions

Both reviewers were very positive about the manuscript. However, both have requested clarification of some points and some additional (although not extensive) analyses. Please address all of the comments from both reviewers in which additions or clarifications are mentioned.

·

Basic reporting

The present research focused on an important topic, which is the estimation of the prevalence of CVS in Italian workers using a validated questionnaire. The authors should be congratulated on their manuscript since to date they are the first who have used a validated questionnaire to measure this syndrome.

This research concludes that sex (to be a woman), regular use of optical correction, work, and to the time of use of digital devices to work increases the odds of suffering CVS. My main concern is that the authors cannot explain if the relationship among the regular use of optical correction and CVS is due to an incorrect subjective refraction or to other reasons like accommodative or binocular dysfunctions. This could have been solved with the measure of visual acuity data from the participants. Nevertheless, as the authors mention, their topic was to investigate CVS with some risk factors related to work characteristics and tear film and ocular surface characteristics. I suggest the authors to include variables as visual acuity or subjective refraction in future research.

Comment 1: Minimal technical corrections should be done in the use of English:

The English language is clear and well-written, although several sentences should be shortened to make reading the article more fluent. Some examples are: a) paragraph in lines 90 to 95 and b) paragraph in lines 132 to 145. Both are too long and it becomes complex to understand. I suggest the authors to divide them in shorter sentences.

Subject-verb agreement should be checked. Some examples where the verb should be corrected and changed are the following:

Line 291: “This result suggest” should be corrected and changed by “This result suggests”

Line 308: “they implies” should be corrected and changed by “they imply”

Line 350: “Uses of a validated questionnaire...” should be corrected and changed by “The use of a validated questionnaire...”

Concerning the spelling:

Line 87: tweenager by teenager

Line 105: separate VDU and workers

Line 142: delete the parenthesis after work, since it is not necessary.
Line 271: number 32 should be written as a superscript.

Experimental design

The methodology is rigorous and appropriate.

Validity of the findings

Tables and figures give a comprehensive breakdown of the results obtained. The discussion follows logically from the results. References are appropriate.

Comment 2: The authors state in lines 273-277 that the association between being female and a greater probability of suffering from visual and ocular symptoms or even CVS could be explained by the fact that 60.1% of women in the study would be over 45 years which means that they probably are perimenopausal or menopausal; so, the reason of the greater prevalence of CVS could be the influence of hormonal factors. It would be interesting to analyze if there are differences in the prevalence of CVS when comparing women under 45 and women over 45 years of age; also considering if the exposure to the computer (hours/day) was the same. These results could help to confirm the hypothesis of the authors.

Reviewer 2 ·

Basic reporting

The authors do an excellent job with the introduction of the paper, it shows good context; there are many relevant citations included. I think the story told in the introduction could be improved to focus in on what is novel and unique to this investigation. For example, lines 80-83 describe exactly the results you find in this study in regards to the prevalence of eye fatigue. Help the reader understand how what you did is different than that; what are you adding to our general knowledge about eye fatigue with this manuscript. Further in lines 105-106 you describe CVS prevalence in Italy; maybe more emphasis on WHY we shouldn’t trust the data from unvalidated questionnaires.
While incredibly thorough, the methods section gets a bit wordy, especially lines 132-142 (maybe include information about what is included in the questionnaires in an appendix?). The CVS-Q is a validated questionnaire (although you should cite the paper it came from) so stating you used it with its criteria for diagnosis is adequate. Similarly, in lines 158-162, you’ve already stated that the criteria from TFOS were used, so you can succinctly state the criteria were from TFOS and they were x,y,z.
The English language is quite good. A few suggested improvements include removal of ‘in’ in like 74, and amending line 199 to read ‘during the day’. The use of the word ‘altered’ beginning in line 158, in regards to classification of ocular surface testing seems strange as well; maybe consider these tests as being ‘normal’ or ‘abnormal’.
In alignment with the Journal criteria, doe the software used (line 127 and 181) need cited or websites listed?
I think you could combine tables 1 and 2 to more succinctly present all of the data. One table with a subheading of full population and CVS population and the included numbers would be more clear. Similarly, tables 3 and 4 could easily be combined and allow easier viewing of the data.

Experimental design

Generally the study design seemed robust, the research question was well stated. The investigation was performed at a high technical and ethical standard
Just a few considerations; I might think that someone who has a history of strabismus (line 190) would not have a normal binocular vision status, thus would be susceptible to similar symptoms as CVS from their binocular vision disorder. Also, while I understand your desire to not include contact lens wearers in your sample, I wonder if this inadvertently skewed your sample to focus on an older population (eg. No subjects 18-23yoa); while there’s not much you can do about that now, maybe acknowledge that in your discussion, or consider this when designing future studies.
With half of your subjects being older than 45 (line 188), you could assume most of them would not have a normal accommodative status. I wonder how this would also impact reports of CVS.

Validity of the findings

Data provided looks robust. Analysis is statistically sound
It is interesting that while you include many presbyopes in your sample, only 17% of your subjects used a progressive lens (line 194, 195). Does this mean that many of the subjects were using single vision lenses to look at the computer screen (e.g. readers)? Maybe it would be more effective to break the monofocal category down into monofocal – distance and monofocal – near to really understand what the lenses are used for.
In the results, you might consider reporting the median age as well as the mean just to understand more about the distribution of ages in the study.

You have a really solid methods and results section, but I think a little work on the conclusions could make the paper stronger, and sell the excellent story that your dataset represents.
Generally, I think you need to include a stronger justification (lines 303-305) for why you did not ensure the subject’s refractive status was optimally corrected before querying them on eye fatigue (particularly when the most frequent symptom was blurred vision, line 211). While I assume this is because you were targeting a more observational cross sectional design, I think we need to be careful to ensure that we aren’t just classifying people who simply need an updated glasses prescription as having eye fatigue.
In line 275, if you’re going to acknowledge perimenopausal or menopausal women as a potential factor in your results, then wouldn’t we expect to see the dry eye testing to be abnormal in that population? Did you see that? You hint at this conclusion, but if you’re going to include it, finish out the full thought of why perimenopausal/menopausal women would be more susceptible to CVS
In line 250, while your results cannot be directly compared, and some of the other studies were different, in your introduction (lines 105-106) you cite a prevalence of 13-88%, so your prevalence of 67.2% would be within that range. I think using what was found previously to support what you found would be good, then explaining why what you did was more robust/valid/relevant to the current times/able to be compared in future studies, etc.
Be careful not to insinuate causation with correlation; particularly digital device use of >6h (lines 286-292. What if there is a third factor that is both causing excessive digital device use and CVS, such as stress levels?
I didn’t follow your argument about the increased probability of CVS among workers using optical correction in lines 305-315. Following your logic, wouldn’t someone who had a corrected refractive problem actually have an easier time focusing on different tasks? I do like that you acknowledge in lines 315-318 that that people with CVS may be more likely to get their eyes checked/get glasses, which is incredibly valid.

Additional comments

Overall, great work! I can tell a lot of effort went into designing, conducting, and analyzing this dataset, and for that I applaud all of the authors. I think if one thing could be done to improve this paper, it would be really thinking about the story you want to tell with the introduction and conclusion, and revising as appropriate. The population you studied was unique in many ways, and your study captured important information using a validated questionnaire. Tell your readers why this data is important.
I think you have a very unique population, if the digital device use in Italy is truly that low, it seems that continued work with this population could yield some very interesting conclusions, when compared to other populations where digital device use is much higher. I wish you the best of luck with your future research endeavors.

---

## Round 0.2 · accepted · Accept

Thank you for responding so thoroughly to all of the reviewer comments. While I'm sure you have done this already, please double-check all of the references and reference numbers.

·

Basic reporting

The authors have carefully answered all the questions raised in the first review. Thank you for clarifying all the questions.

Experimental design

No comment

Validity of the findings

No comment

Additional comments

No comment